# TANGRAM: A DATASET FOR FPGA-BASED HETEROGENEOUS SYSTEMS-ON-CHIP OPTIMIZATION

## ABSTRACT

With the end of Moore's Law and Dennard Scaling, high-performance computing (HPC) architectures are evolving to include large Field Programmable Gate Arrays (FPGAs) to improve efficiency. Identifying the optimal configuration for such FPGAs, in terms of the number and type of CPUs, hardware accelerators, and memory channels, is crucial for the creation of efficient computing platforms. However, the complexity of the design space, the difficulty of modeling the interactions between the concurrently executed applications, and the strict time-to-market requirements fostered the use of heuristics to perform the exploration, thus leading to the identification of suboptimal solutions with no quality guarantees. To support the exploration of new systematic methodologies for the design of FPGA-based heterogeneous multi-core architectures, we present TANGRAM, a dataset composed of performance and resource consumption results of more than $40,000$ different designs, collected from two high-end FPGAs executing heterogeneous and concurrent applications. To assess the suitability of this dataset for machine-learning-based optimization strategies, we tested it with some baseline regression methodologies, showing the possibility of accurately predicting the performance of multiple applications running on the same system.

## 1 INTRODUCTION

With the decline of Moore's Law (Moore, 1998) and Dennard Scaling (Dennard et al., 2018), hardware acceleration has become the de facto standard in data centers, with service providers continuously upgrading their infrastructures to meet the growing computational demands of modern applications (Theis and Wong, 2017). In this scenario, FPGAs have seen widespread adoption by major companies like Microsoft, AWS, Alibaba, and Huawei due to their superior efficiency compared to traditional CPU- and GPU-based architectures (Caulfield et al., 2016). High-end FPGAs now incorporate millions of logic elements and large on-chip memories, making them capable of handling complex and heterogeneous computing platforms while concurrently implementing a wide range of hardware accelerators. For instance, an Advanced Encryption Standard (AES) accelerator (Joachim Strömbergson, 2014) implemented on an AWS F1 instance with an AMD Virtex UltraScale+ FPGA uses less than 0.4% of the available programmable logic cells. Considering this large resource availability and the significant cost of high-end FPGAs, dedicating an entire device to a single application represents a suboptimal approach, both computationally and economically. This consideration has fueled the development of several strategies that allow sharing an FPGA between multiple applications, like multi-tenancy frameworks (Bobda et al., 2022).

Such an approach, however, poses the problem of identifying the optimal application allocation on an FPGA to satisfy a set of service-level agreements (SLAs) between the users and the provider while minimizing the resources utilized by the design to reduce the provider costs. There are primarily two factors that make this problem extremely complex. First, the number of possible FPGA configurations increases exponentially with the number of applications and the resource availability on the device, making an exhaustive exploration impractical. Second, the interaction between several applications sharing the same resources, especially interconnect buses and memory connections, represents a problem that cannot be solved in a closed form. For this reason, several heuristics have been proposed to approach the identification of the optimal FPGA configuration (Jordan et al., 2021; Damak et al., 2014; Brandalero et al., 2019; Xu et al., 2019; Cong and Charot, 2021). However, they require

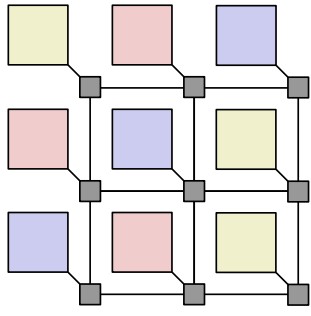

Figure 1: A schematic representation of a 3×3 SoC. The colored squares represent the tiles, while the black grid is the NoC.

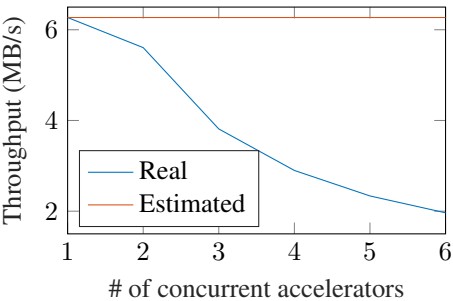

Figure 2: Real and estimated throughput of a *dfadd* accelerator with different numbers of accelerators sharing the same memory.

in-depth knowledge of the specific system they try to model, failing to provide a general solution that can be applied to any FPGA architecture or application set.

This paper presents TANGRAM, a dataset containing the area and performance statistics of more than $40,000$ heterogeneous systems-on-chip (SoCs), where several hardware accelerators run concurrently, sharing the same interconnect and memory resources. TANGRAM is composed of high-quality data collected through the implementation and execution of the designs on two different HPC-oriented FPGA boards. The goal of TANGRAM is to provide a set of ready-to-use data that can be employed to develop novel machine-learning techniques targeting the optimization of heterogeneous SoCs. In fact, the adoption of machine learning is crucial for identifying a general solution to this problem, overcoming the need for low-level knowledge characterizing the use of heuristics.

We validated TANGRAM using classical machine-learning algorithms (Williams and Rasmussen, 2006; Segal, 2004; Peterson, 2009), solving two different problems: regression and optimization. The former allowed us to demonstrate the possibility of performing accurate predictions regarding the throughput of concurrently executing hardware accelerators with minimal a priori knowledge. For the latter, we combined the predictive features of machine learning models with a multi-armed bandit approach to identify the optimal configuration given a specific set of constraints. The dataset is completely open source and publicly available at [1].

## 2 BACKGROUND

A *system-on-chip* is an integrated circuit that implements a computing platform made of an on-chip interconnect, a set of computing elements, a set of memories, and auxiliary and I/O logic. An SoC is defined *heterogeneous* if it implements computing elements of different kinds, for example, general-purpose CPUs and domain-specific accelerators. The on-chip interconnect of modern SoCs is usually implemented with a network-on-chip (NoC) (Bjerregaard and Mahadevan, 2006), which is composed of routers, each of them associated with a *tile* performing a specific functional role, connected together with links. Without losing generality, we consider 2D-mesh topologies (as the one shown in Figure 1), since it is the most widespread implementation of NoCs (Bjerregaard and Mahadevan, 2006).

In this context, we define a *configuration* of a SoC as a specific arrangement of tiles over the NoC grid. Configurations differ for the size of the NoC and for the type and implementation of the tiles. Generally speaking, SoCs are evaluated using two main metrics, resource consumption and performance. The resource consumption represents the area occupied by the integrated circuit, and it is a direct indicator of its total implementation cost. On FPGAs, a SoC using more resources may require a larger device to host it. However, the area of a SoC can be estimated with a good degree of accuracy as the sum of the areas of its tiles and its interconnect extracted in isolation, so it can be assessed in advance before implementing the SoC on the FPGA.

---

[1]The url is omitted for blind review.

On the contrary, the performance of different computing elements sharing resources (in particular, the interconnect and the memories) is much more difficult to predict, since it depends on the specific hardware implementation of the various components included in the SoC. This is exemplified in Figure 2, which depicts the throughput of a *dfadd* accelerator (Hara et al., 2008) instantiated on a 2D-mesh architecture featuring a single memory channel, implemented on an AMD Virtex7-2000 FPGA. The actual throughput of the accelerator decreases when multiple accelerators are running concurrently, with an error of almost 70% with respect to a naive estimation of the performance ignoring the effect of resource sharing.

Thus, the performance of a SoC is tightly related to its configuration, and its increase is often achieved at the expense of more area consumption. For example, a computing element can be parallelized within a tile or spanned over several tiles to improve its performance. Alternatively, more tiles can be devoted to memory channels, sacrificing raw computing power for a higher communication bandwidth (such an approach would be particularly effective in the scenario depicted in Figure 2, where the memory access constitutes the performance bottleneck). In any case, the identification of feasible configurations, i.e., the ones achieving the desired area-performance trade-off, is a complex problem requiring an extensive design space exploration (DSE) of the possible configurations.

Indeed, exploring the configuration space is a challenging goal, for two main reasons. The first issue is the huge size of the design space, which increases exponentially with the number of tiles in the SoC and the available functionalities that can be hosted by each tile. The second problem is the time required to assess the performance of a configuration. Whether this process employs simulations (whose accuracy, however, is suboptimal (O'Neal and Brisk, 2018)) or implements the design on a real FPGA, the evaluation of a single configuration may require tens of minutes, or even hours, setting a limit on the maximum amount of configurations that can be effectively tested.

## 3 RELATED WORKS

Several studies in the literature have addressed the optimization of heterogeneous SoCs as a DSE problem. For example, Damak et al. (2014) formulates the design space exploration as a mixed-integer linear programming (MILP) problem, whereas Brandalero et al. (2019) devises an application-specific metric to optimize the assignment of shared accelerators to CPU cores. To further reduce exploration time, Xu et al. (2019) introduce a two-phase methodology that separates configuration generation from SoC evaluation. The evaluation step can rely on either cycle-accurate simulation, approximate simulation with accelerator models, or direct execution on FPGA hardware. Jordan et al. (2021) propose an algorithm based on the *genetic multidimensional knapsack* approach, that can swiftly determine the optimal task deployment on a heterogeneous system. The work of Cong and Charot (2021) presents an iterative design exploration method that continuously refines a hyperparameter-based performance model using feedback from previously evaluated configurations. This strategy allows for the progressive improvement of design quality during exploration.

Notably, all of the discussed proposals adopt heuristics-based approaches, which need a deep knowledge of the tasks to allocate and are not generalizable to different types of SoCs (Kim et al., 2018). Conversely, the adoption of machine learning would allow the extrapolation of an accurate performance model of an SoC using execution statistics from previously evaluated configurations, providing a general approach that can be easily extended to any SoC architecture or application set. We mention that machine learning was successfully employed in other aspects of the design and optimization of SoCs, including the enhancement of accelerators generated with high-level synthesis (HLS) (Wu et al., 2023)(Bai et al., 2023), the floorplanning on the FPGA (Farooq et al., 2021), and the design of the NoC (Rao et al., 2018). We believe that the lack of machine-learning-based solutions to the problem of accelerators' performance prediction in heterogeneous SoCs is partially caused by the unavailability of a large and curated dataset of SoC execution statistics since collecting results for large systems requires a huge amount of computational time together with specialized hardware design skills.

## 4 THE TANGRAM DATASET

TANGRAM is a curated dataset containing the resource consumption and throughput statistics of more than 40, 000 different heterogeneous SoCs. This massive amount of data is the result of several

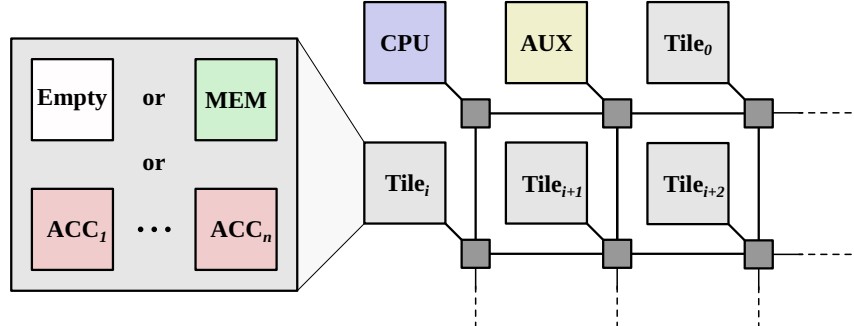

Figure 3: Architecture of the SoC employed for the collection of TANGRAM results.

months of computation, employing two different HPC-class FPGA boards for the collection of real execution statistics. The rest of this section is divided into three parts. Section 4.1 details the experimental setup employed for data collection. Section 4.2 describes the content of the TANGRAM dataset. Finally, Section 4.3 explains the format used to store the execution and area results of the SoCs.

## 4.1 EXPERIMENTAL SETUP

**Hardware and software setup** All the results available in the TANGRAM dataset have been collected through the execution on FPGA boards (thus, no simulations or modeling have been used). The experiments targeted two different devices belonging to distinct FPGA families, with the goal of providing a more varied set of results. The first board is part of a Siemens proFPGA prototyping system, consisting of an FPGA daughter board (*fm-xctv2000t-r2*) (Siemens Electronic Design Automation GmbH, a), that features an AMD Virtex-7 2000T FPGA (*xc7v2000t-2flg1925-c*), mounted on a quad motherboard (*mb-4m-r3*) (Siemens Electronic Design Automation GmbH, b). With 1221600 look-up tables (LUT), 2443200 flip-flops (FF), 2584 18Kb blocks of block RAM (BRAM), and 2160 digital signal processing (DSP) elements, the Virtex-7 2000T is the largest device of AMD's Series 7 family. The second board is an AMD Alveo U55C, which is based on an FPGA belonging to the AMD Virtex UltraScale+ family (*xcu55c-fsvh2892-2L-e*). This device features 1303680 look-up tables (LUT), 2607360 flip-flops (FF), 4032 18Kb blocks of block RAM (BRAM), and 9024 digital signal processing (DSP) elements. Moreover, the U55C chip is equipped with 16GB of 2nd generation high-bandwidth memory (HBM2).

The software used to compute and collect the data was AMD Vivado 2019.2 for synthesis, implementation, and bitstream generation, Siemens proFPGA Builder 2019A-SP2 for FPGA programming, and AMD Vivado HLS 2019.2 for the HLS of accelerators. The host computer executing this software was equipped with an Intel i5 processor (*Intel Core i5-12400*) and 64GB of DDR4 RAM, with a Kubuntu 22.04 OS.

**Architectural setup** TANGRAM makes use of NoC-based SoCs, generated using the open-source SoC prototyping platform described in (Montanaro et al., 2024), in which each tile can assume any implementation between CPU, AUX, MEM, and ACC, or can remain empty, as shown in Figure 3. Every SoC allocates just one tile for the CPU (a CVA6 64-bit RISC-V processor (Zaruba and Benini, 2019)) and another one for the auxiliary functions, which are always placed in the top-left corner of the mesh. Notably, the CPU and AUX tiles have little to no impact on the accelerators' performance since they produce very limited traffic on the NoC. Thus we decided to fix their position in the top-left corner to avoid an increase in the (already huge) design space. As for the other tiles, they can assume any other implementation (i.e., MEM, ACC, or remain empty) in no particular order.

Concerning the memory tiles, each of them is completely independent from the others, ensuring the absence of any interference between accelerators allocated to different memory tiles. Moreover, in our experiments we have employed two distinct on-chip memory resources: BRAMs and HBM2 (the latter is only available on the Virtex UltraScale+ FPGA). The memory implementations that have been chosen for each scenario are specified in Tables 1 and 2.

Table 1: List of the exhaustive results included in the TANGRAM dataset. Legend: **UC** use case, **EX** exhaustive, **Mem.** memory implementation, **disp.** dispositions, **comb.** combinations, **#Conf.** number of configurations.

| UC | FPGA | Mem. | NoC | Space size | Application set | Strategy | #Conf. |
|----|------|------|-----|-----------|-----------------|----------|--------|
| EX1 | Virtex 7 | BRAM | 2×3 | 342 | *dfadd* | disp. | **342** |
| EX2 | Virtex 7 | BRAM | 2×3 | 972 | *adpcm, dfadd* | disp. | **972** |
| EX3 | Virtex 7 | BRAM | 3×3 | 157 | *adpcm* | comb. | **157** |
| | | | | | *aes* | comb. | **157** |
| | | | | | *dfadd* | comb. | **157** |
| | | | | | *dfmul* | comb. | **157** |
| | | | | | *gsm* | comb. | **157** |
| | | | | | *sha3* | comb. | **157** |
| EX4 | Virtex US+ | HBM2 | 3×3 | 3294 | *aes, dfadd, motion* | comb. | **3294** |
| **Total** | | | | | | | **5550** |

Regarding the accelerator tiles, they can be implemented with different parallelism levels in order to increase their throughput without occupying more tiles. Most of the employed benchmarking applications (*adpcm*, *dfadd*, *dfdiv*, *dfmul*, *gsm*, *mips*, and *motion*) have been taken from the CHStone benchmark suite (Hara et al., 2008), a set of accelerators written in C and generated using HLS. To provide more variability to the results, we added to this set two high-performance cryptographic accelerators (*aes* and *sha3*) written in SystemVerilog (adapted from Joachim Strömbergson (2014) and Josh Moles (2013), respectively).

**On-board execution**  All area results have been collected after the implementation phase of the SoC in Vivado. Regarding the collection of the execution statistics, it has been performed as follows. First, the bitstream is sent to the board and the SoC is instantiated on the FPGA. Then, a binary file containing the testing program is loaded into the main memory and the system is booted. Finally, the program starts on the CPU tile: during its execution, it initiates all the accelerators, lets them operate for several seconds, and sends the execution statistics to the host server through a USB-to-serial interface.

### 4.2 DATASET DESCRIPTION

Tables 1 and 2 list all the use cases included in the dataset, with the number of collected configurations for each scenario. The use cases differ for the employed FPGA (the Virtex 7 included in the ProFPGA Virtex-7 2000T board or the Virtex UltraScale+ featured in the Alveo U55C), for the chosen memory implementation (BRAM or HBM2), for the size of the NoC, and the application set. The use cases NX7 to NX17 allow the accelerator tiles to assume two levels of parallelism (1 and 2), while the rest of the scenarios permit also a third level of parallelism (4). Depending on the setting, we provided either an exhaustive (EX) or a non-exhaustive (NX) search of the configuration space.

More specifically, we performed an exhaustive exploration only for scenarios with a limited design space size ($< 5000$). They are an example of those results that are obtainable in a reasonable amount of time during the design phase of a SoC. The EX1 and EX2 experiments involve a smaller tile configuration ($2 \times 3$), with one and two applications, respectively. Instead, EX3 and EX4 evaluated larger SoCs ($3 \times 3$), consequently increasing the configuration space. Notably, during the collection of EX1 and EX2 results, we found only small discrepancies in terms of performances between configurations having the same accelerators and memories displaced differently over the tiles. For this reason, we decided to consider only the possible accelerators and memory combinations for the larger scenarios, reducing the total amount of configurations with respect to the space of all the possible dispositions.

The non-exhaustive searches represent those scenarios in which it was not possible to perform the complete exploration in a reasonable time. Such settings simulate a realistic scenario in which the huge size of the design space prevents the identification of the globally-optimal configuration, and thus,

Table 2: List of the results of non-exhaustive explorations included in the TANGRAM dataset. Legend: **UC** use case, **NX** non-exhaustive, **Mem.** memory implementation, **#Conf.** number of configurations.

| UC | FPGA | Mem. | NoC | Space size | Application set | #Conf. |
|----|------|------|-----|-----------|-----------------|--------|
| NX1 | | | | 52 432 | *adpcm, aes, dfadd* | **461** |
| NX2 | | | | | *adpcm, dfadd, sha3* | **426** |
| NX3 | Virtex 7 | BRAM | 3×4 | 178 443 | *adpcm, aes, dfadd, gsm* | **406** |
| NX4 | | | | | *adpcm, dfmul, gsm, sha3* | **476** |
| NX5 | | | | 352 134 | *adpcm, aes, dfadd, gsm, sha3* | **461** |
| NX6 | | | | | *adpcm, aes, dfadd, dfmul, gsm* | **401** |
| NX7 | | | | | *aes, dfadd, sha3* | **4 111** |
| NX8 | | | | 18 578 | *aes, dfadd,motion* | **1 215** |
| NX9 | | | | | *dfadd, dfdiv, dfmul* | **6 858** |
| NX10 | | | | | *aes, dfadd, motion, sha3* | **358** |
| NX11 | | | | 79 740 | *aes, mips, motion, sha3* | **208** |
| NX12 | Virtex US+ | BRAM | 4×4 | | *dfadd, dfdiv, dfmul, motion* | **5 649** |
| NX13 | | | | | *aes, dfmul, mips, sha3* | **4 211** |
| NX14 | | | | | *aes, dfdiv, dfmul, motion, sha3* | **767** |
| NX15 | | | | 221 055 | *aes, dfadd, dfmul, mips, sha3* | **4 483** |
| NX16 | | | | | *dfadd, dfdiv, dfmul, mips, motion* | **2 474** |
| NX17 | | | | | *aes, dfadd, dfdiv, dfmul, motion* | **1 833** |
| NX18 | | | | 308 813 | *aes, dfadd, motion* | **799** |
| NX19 | Virtex US+ | HBM2 | 4×4 | 1 962 040 | *aes, dfadd, dfdiv, motion* | **802** |
| NX20 | | | | 7 499 673 | *aes, dfadd, dfdiv, dfmul, motion* | **791** |
| **Total** | | | | | | **37 210** |

it is necessary to employ modeling and optimization techniques to efficiently explore the design space and find the best SoC configuration. Fostering the research of novel SoC optimization techniques is, in fact, the main goal of the TANGRAM dataset. The selection of the configurations included in the NX scenarios have been performed in part at random (over the whole design space), and in part by using a sequential decision-making approach (i.e., multi-armed bandit), whose implementation is detailed in the Appendix.

Listing 1: Dataset entry extracted from EX2.

```
1  "ADPCMx1_ADPCMx1_DFADDx2_MEM": {
2      "BRAM18": 1212, "DSP": 207, "FF": 93558, "LUT": 100584,
3      "throughput": {"ADPCM_0": 1.375, "ADPCM_1": 1.374,
4          "DFADD_0": 12.011},
5      "throughput_total": {"ADPCM": 2.749, "DFADD": 12.011},
6      "tile_config": {
7          "X0Y0": "CPU", "X0Y1": "ADPCMx1", "X0Y2": "DFADDx2",
8          "X1Y0": "AUX", "X1Y1": "ADPCMx1", "X1Y2": "MEM"}
9  }
```

## 4.3 DATA FORMAT

In Listing 1, we report an entry of the dataset formatted as a JSON file. The information provided for each entry is divided into three main parts: area, performance, and tile configuration. Four different fields are used to express the area of a configuration: `BRAM18`, `DSP`, `FF`, and `LUT`, representing the number of 18Kb BRAM blocks, DSP elements, FFs, and LUTs, consumed on the device by the complete system, respectively. The performance is further subdivided into two fields: the throughput of the single accelerator tiles (`throughput`) and the global throughput of the application (`throughput_total`). The accelerator tiles are identified by a number, that indicates the appearance order of that accelerator tile in the SoC. For instance, in the example, we have a

throughput of 1.375 for the first *adpcm* accelerator listed in the configuration name, 1.374 for the second *adpcm* one, and 12.011 for the *dfadd* one. Instead, the global application throughput is the sum of the individual throughput of the accelerator tiles. Regarding the configuration section, each tile is uniquely identified by its X-Y coordinates. Non-accelerator tiles can assume 4 different implementations: CPU, AUX, MEM, or EMPTY. Accelerator tiles are characterized by the specific application name, and by the parallelism level of the accelerator (the figure after the x character in the configuration name).

## 5 EXPERIMENTAL EVALUATION

To verify the potential of the TANGRAM dataset for the development of new prediction and optimization strategies targeting complex heterogeneous SoCs, we carried out an experimental campaign to show how different machine-learning models can make use of this testbed. In particular, we formulate two different problems: a regression problem, targeting the estimation of the throughput of a specific configuration, and an optimization problem, which has the goal of identifying the optimal configuration library starting from a minimal set of information regarding the SoC and the accelerators using a multi-armed bandit strategy to efficiently explore the configuration space. In the rest of this section, we expand the description of these problems and provide the experimental results obtained by applying these approaches to the TANGRAM dataset.

### 5.1 PROBLEM DEFINITION

We cast the problem in two different flavors: as a classical regression and as a multi-armed bandit problem.

**Regression**  Given a set of configurations for which we already have performance information, the goal is to estimate the overall throughput of each application over the same architecture for any newly provided configuration. This scenario can be cast as a classical multi-output regression problem, where a subset of the information available for each accelerator in a given configuration is used as input. For our specific model, the features employed to model the accelerator's behavior are the parallelism level of the tile itself and the number of memory- and compute-bound accelerators allocated on the same memory tile.[2] Here, we considered different standard approaches for solving this problem, as Gaussian Process (Williams and Rasmussen, 2006), Random Forest (Segal, 2004), and K-Nearest Neighbors (Peterson, 2009). In this setting, it is crucial to quantify the minimal amount of data required to get a good throughput approximation, while minimizing the number of implemented configurations to reduce exploration time. In order to answer this question, we perform estimations using different percentages of the design space for the training of the regression models in the exhaustive scenarios, progressively increasing the amount of training data.

**Multi-armed bandit**  Given a set of constraints in terms of application throughput (i.e., a set of SLAs), the goal of this optimization problem is to minimize the total area of the SoC. This problem can be cast as a combinatorial bandit problem (Chen et al., 2013), in which the arms correspond to the different accelerators, the set of the feasible configuration is provided by the thresholds, and the objective function is the overall area utilization. In our case, we use the number of LUTs as a proxy for the overall area consumption of the design. We resorted to the approach provided in Accabi et al. (2018) to solve such a problem. We compare this FPGA combinatorial multi-armed bandit (FC-MAB) approach with a random strategy that selects configurations at random. This problem can be applied only to exhaustive scenarios since in non-exhaustive settings, the FC-MAB selector would be limited in the choice of which configurations to explore.[3] Moreover, the availability of all the possible configurations allows us to match the optimization strategy against the absolute optimal configuration.

**Performance Evaluations and Parameters**  To implement the regressors, we used the *scikit-learn* Python library functions with default parameters. For the FC-MAB, we set the exploration parameter

---

[2]We provided in the Appendix a detailed mathematical description of the employed model.

[3]Notice that comparing such methods with other heuristics mentioned in Section 3, would require to have a priori information about the setting. Conversely, a bandit approach can be run starting from scratch.

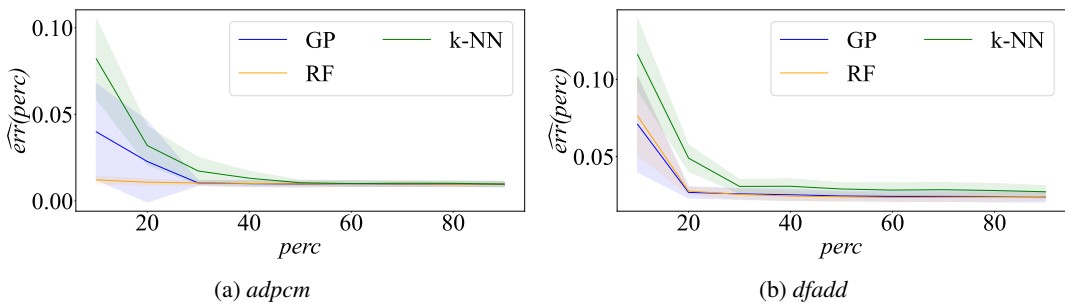

Figure 4: Regression error for two different applications in the EX3 scenario.

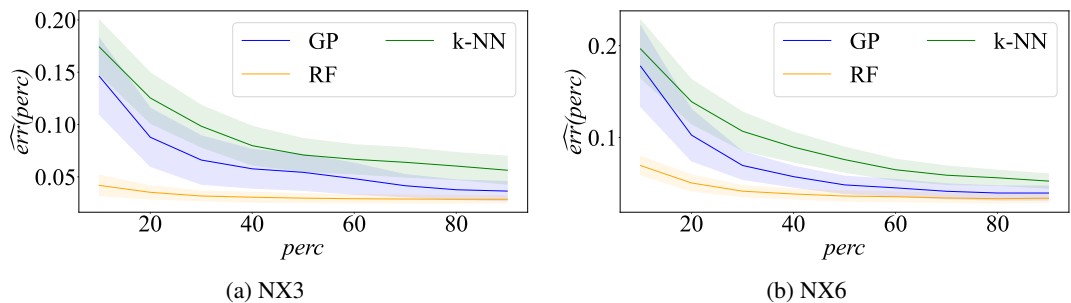

Figure 5: Regression error in two different multi-application scenarios, as the average error of the single applications.

for FC-MAB $b_{j,t-1} = 5$ and we run it over 50 iterations. We compared the analyzed algorithms in terms of:

- $\widehat{err}(perc)$ [adimensional] normalized average reconstruction error for the throughput of a specific accelerator type using a percentage $perc$ of the available configurations as the training set (averaged over the configurations and the instances of the same accelerator present in each configuration);

- $\widehat{thr}(n)$ [MB/s] average throughput after $n$ rounds of the configuration selected by the algorithm;

- $\widehat{LUT}(n)$ [adimensional] occupied area of the configuration selected by the algorithm;

where the average is taken with 10-fold cross-validation in the regression problem and w.r.t. 30 independent runs of the algorithms in the multi-armed bandit case. We report the 95% (Gaussian) confidence intervals in the plots as semi-transparent areas.

## 5.2 EXPERIMENTAL RESULTS: REGRESSION

Figure 4 shows the regression error $\widehat{err}(perc)$ of the employed model for the *adpcm* and *dfadd* applications in the EX3 scenario. Figure 5, instead, portrays the average regression error over the application set for the NX3 and NX6 settings. A larger set of regression errors collected in our experiments is provided in the Appendix. These examples show the ability of the models to predict the throughput of the applications with an acceptable degree of accuracy. As expected, larger training percentages provide a significant reduction in the overall error. In particular, we can observe two different behaviors when comparing EX3 scenarios in Figure 4 with larger and more complex ones in Figure 5. The former reaches the minimum error when using $20-30\%$ of the available configurations as a training set, while the latter shows a continuously improving trend proportional to the training sample percentages. This behavior can be explained by considering that the EX3 dataset is exhaustive, while the non-exhaustive scenarios include just a small fraction of the design space ($\approx 0.2\%$). Indeed, the ability to reach an accuracy lower than the $5\%$ with such a minimal exploration of the design space represents a promising result. As a final consideration, the k-NN model always returns the highest

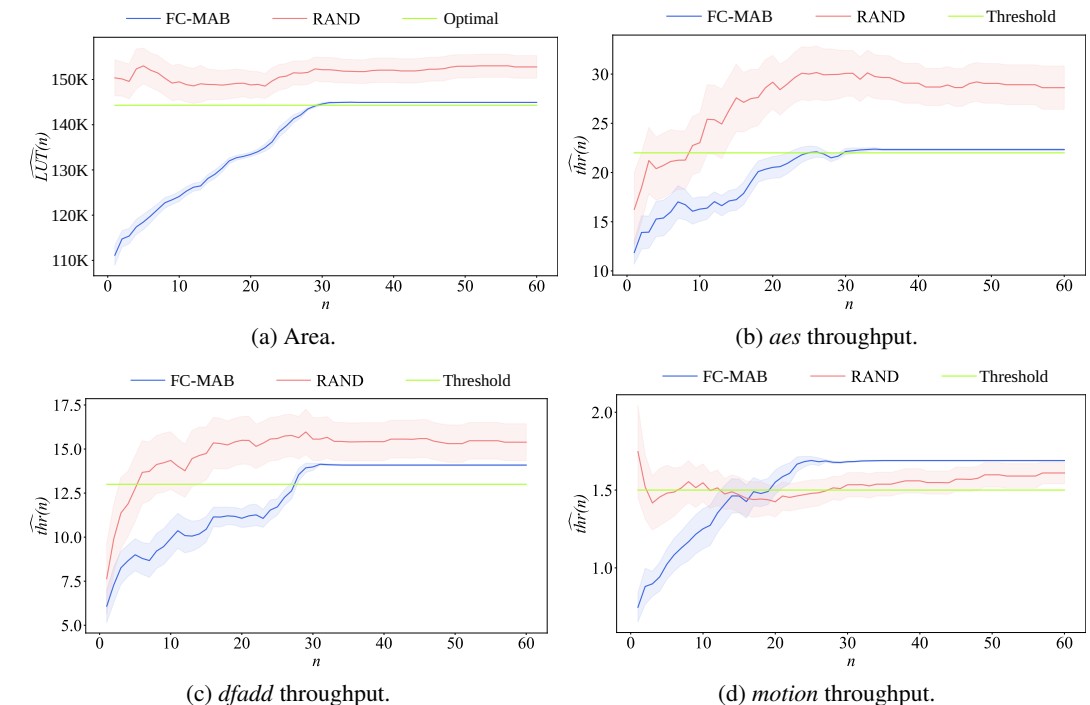

Figure 6: Area and throughput of the best-identified configuration at each iteration of the optimization flow for the EX4 scenario.

error, with RF usually achieving the best accuracy. These results confirm the idea that machine learning models can prove extremely useful in predicting the performance of multiple accelerators interacting on the same system.

### 5.3 Experimental Results: Multi-Armed Bandit

Figure 6 provides the evolution of the throughput and the area consumption over the rounds for the FC-MAB strategy targeting the EX4 scenario. Similar results have been obtained by running the optimization process over the other three exhaustive scenarios. In all the studied cases, the FC-MAB was able to converge to the optimal solution before the end of the 50 rounds. For the EX4 scenario, this allowed us to identify the optimal solution by exploring just the 1.5% of the design space. Conversely, the random approach is unable to consistently identify configurations that satisfy the throughput constraints: over the 30 runs performed for the EX4 scenario, it found a feasible configuration just 14 times. These promising results suggest that the use of online learning approaches for the optimization of heterogeneous SoCs constitutes a suitable option.

## 6 Conclusions

This paper presented TANGRAM, a dataset comprising the area and execution statistics of more than 40,000 heterogeneous SoCs collected on datacenter-grade FPGAs. For each configuration included in the dataset, we reported the area and performance results as well as the allocation of each application. The statistics have been extracted by implementing and executing the corresponding configuration on real FPGA boards, namely a Siemens ProFPGA Virtex-7 2000T and an AMD Alveo U55C, which required several months of computation. An extensive set of experimental results demonstrates the possibility of accurately modeling complex SoCs with different regression strategies, motivating the need to explore the use of machine learning for the optimization of heterogeneous SoCs. The TANGRAM project is still under development, and we plan to expand the dataset with new results and use cases.

## REPRODUCIBILITY STATEMENT

The regression and optimization results reported in this study were derived from the SoC area and performance statistics provided in the dataset. These results are fully reproducible using the source code available in the dataset repository, which is documented both within the repository and in the Appendix.

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

# APPENDIX

## A TANGRAM DATASET DESCRIPTION

In this section, we describe the content and the main features of the TANGRAM dataset. In particular, Section A.1 details the organization of the TANGRAM repository, while Section A.2 outlines its expected usage.

### A.1 DIRECTORY CONTENT

Figure 7 shows the structure of the TANGRAM folder. The root directory contains a `README.md` file, which briefly introduces the dataset and explains how to replicate the experiments described in Section 5, a `LICENSE` file providing the Apache 2.0 license of the dataset, and three sub-folders: `data`, `models`, and `scripts`. The `data` folder contains all the area and performance results related to the SoCs implemented and tested during this study. The results are divided into two folders, `EX` and `NX`, following the organization described in Tables 1 and 2, respectively. Each dataset file is characterized by a prefix and a suffix. The prefix indicates the name of the corresponding scenario. Instead, the suffix specifies the strategy employed to collect the data. There are three possible suffixes: *opt*, meaning that the data were collected through an optimization process, *rand*, meaning that the configurations were selected at random, and *total*, which signals that this dataset comprises all the configurations included in the other datasets related to the same scenario. Instead, the `tiles_x.json` files collect some information about the SoC tiles, in particular, the area consumption of non-accelerator tiles and the area consumption with several other features for the accelerator tiles, for the three hardware targets adopted in this study (as described in Section 4.1). The `models` directory includes the three models used to validate the TANGRAM dataset, namely the Gaussian Process (Williams and Rasmussen, 2006), the K-Nearest Neighbors (Peterson, 2009), and the Random Forest (Segal, 2004). The relative

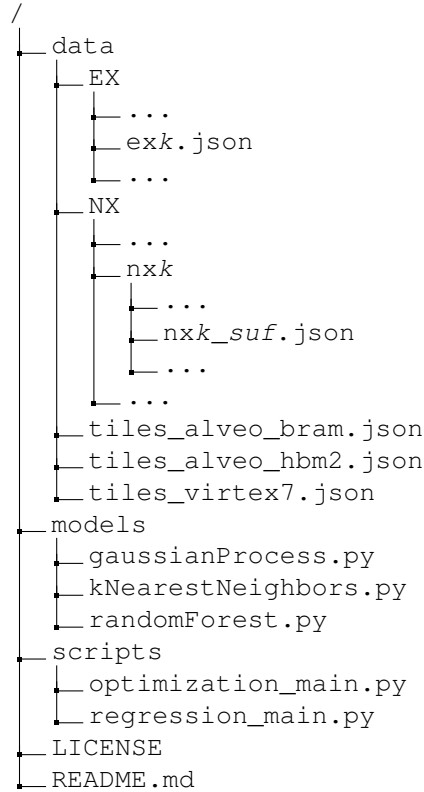

Figure 7: Organization of the TANGRAM repository. The letter $k$ marks the scenario index, while the suffix *suf* represents the strategy used to collect the data.

```
/
├── data
│   ├── EX
│   │   ├── ...
│   │   ├── exk.json
│   │   └── ...
│   ├── NX
│   │   ├── ...
│   │   ├── nxk
│   │   │   ├── ...
│   │   │   ├── nxk_suf.json
│   │   │   └── ...
│   │   └── ...
│   ├── tiles_alveo_bram.json
│   ├── tiles_alveo_hbm2.json
│   └── tiles_virtex7.json
├── models
│   ├── gaussianProcess.py
│   ├── kNearestNeighbors.py
│   └── randomForest.py
├── scripts
│   ├── optimization_main.py
│   └── regression_main.py
├── LICENSE
└── README.md
```

Python sources define various methods that implement the modeling of the problem (that will be detailed later in Section B.1) and the training and throughput prediction functionalities of the models. Finally, the `scripts` folder includes two python scripts, `regression_main.py` and `optimization_main.py`, that execute the experiments described in Section 5.2 and 5.3, respectively. These scripts convert the dataset from JSON to Python dictionaries and then employ the methods defined in the `models` folder to perform the prediction and compute the regression error (`regression_main.py`) or to identify the optimal configuration with an iterative procedure given a set of constraints on the throughput (`optimization_main.py`).

### A.2 INTENDED USE

The goal of the TANGRAM dataset is to allow researchers to develop methodologies for performance prediction and optimization of complex SoCs without having to generate the necessary hardware

results by themselves. In this sense, the scripts included in the repository can be viewed as templates for the use and manipulation of the dataset. In fact, the throughput modeling that we propose, while effectively showing the suitability of TANGRAM for regression purposes, is excessively simplistic and returns a non-negligible error for the more complex scenarios. A more refined methodology could adopt customized regression models and use a more extensive set of accelerators and SoC features to model the applications' throughput. For this reason, we want to encourage researchers to approach the problem with novel ideas and proposals.

# B  METHODOLOGY AND RESULTS

This section details the model used to assess the suitability of the dataset for machine learning methodologies and reports regression results for all the considered scenarios. In particular, the experiments described in Section 5 adopt a specific approach to solve the allocation problem, that makes use of a subset of the features of each accelerator to model the variations in the throughput. These features are then fed to a regression methodology to predict the performance of the accelerators. Moreover, we will provide all the regression results concerning the experiments described in Section 5.2.

The rest of this section is organized as follows. Section B.1 details the mathematical formulation of the allocation setting, as well as the modeling methodology adopted in the dataset validation, while Section B.2 shows the tables containing the complete regression results.

## B.1  SETTING FORMULATION AND REGRESSION STRATEGY

Let us assume that we have a set of applications $P = \{p_i\}_{i=1}^N$. Considering the architecture depicted in Figure 3, the SoC is composed by a fixed number of tiles $T$, where the number of tiles is strictly larger than the number of applications, i.e., $T > N$. Each tile $t_i$, where $i \in \{1, \ldots, T\}$, can host either *i)* an accelerator executing a specific application $p_i$, *ii)* a memory channel, or *iii)* can be left empty. The total throughput can be increased by instantiating multiple accelerators executing the same application on different tiles. Each accelerator can also be implemented with different levels of parallelism $r \in \{0, \ldots, R_{\max}\}$, being $R_{\max}$ the maximum level of parallelism that can be deployed on a single tile.

We model each accelerator $\mathbf{x}_j$ as a tuple $\mathbf{x}_j = \left(r_j, o_j^{(mb)}, o_j^{(cb)}, s_j, a_j\right)$, where $r_j$ is the parallelization of that accelerator on tile $t_j$, $o_j^{(mb)}$ and $o_j^{(cb)}$ are the amount of memory- and compute-bound accelerators allocated to the same memory tile as $\mathbf{x}_j$, respectively, $s_j \in P$ is the application implemented by the accelerator, and $a_j$ is the area consumption of tile $t_j$. A configuration $c = (M, \mathcal{A})$ is defined as the number of memory channels $M$ and the set of accelerators $\mathcal{A} = \{\mathbf{x}_j\}_j$ instantiated on the SoC. The execution of a configuration $c$ returns a vector $\boldsymbol{\theta}(c) = (\theta_1(\mathbf{x}_1), \ldots, \theta_{|\mathcal{A}|}(\mathbf{x}_{|\mathcal{A}|}))$ whose elements $\theta_j(\mathbf{x}_j)$ are the throughput corresponding to the accelerators $\mathbf{x}_j$, where $|\cdot|$ is the cardinality operator.

Generally speaking, given a set of pairs $(c, \boldsymbol{\theta}(c))$, our goal is to predict the global throughput of each application for a generic configuration $c'$, which is not part of the original set. We approach this scenario as a multi-output regression problem, where the input is a subset of the features $\mathbf{z}_j$ of each accelerator $\mathbf{x}_j$ in a configuration $c$. In particular, we model such a relationship using a vector $\mathbf{z}_j := (r_j, o_j^{(mb)}, o_j^{(cb)})$. Given a set of size $k - 1$ of the characteristics $\{\mathbf{z}_h\}_{h=1}^{k-1}$ for a specific accelerator $\{\mathbf{x}_h\}_{h=1}^{k-1}$ each of which is of type $s_j$ and the corresponding throughput $\{\theta_h(\mathbf{x}_h)\}_{h=1}^{k-1}$, the regression model provides an estimate for the expected value of the throughput $\hat{\mu}_{j,k-1}(\mathbf{z})$ of a generic characteristic $\mathbf{z}$. Such a modeling allows to handle the problem with standard regression approaches, like the ones we used for the dataset validation. This modeling, coupled with the FC-MAB strategy described in Section 5.1, was also employed for the optimization experiments reported in Section 5.3.

## B.2  COMPLETE REGRESSION RESULTS

In the following tables, we collected the normalized regression error and the confidence interval for every scenario. The methodology behind these experiments is described in Section 5.2. For each model, we report the regression error and the confidence interval for training set sizes that range from the 10% to the 90% of the entire dataset, whose size is reported in Table 1 and 2.

## C   LLM USAGE

LLMs were used in the preparation of this paper solely as writing assistant tools, specifically to check grammar and improve the clarity of the text.

Table 3: Normalized regression error and confidence interval for all the exhaustive scenarios. Since precision has been set to 0.01, results lower than this figure are shown as 0.0.

| EX | Model | Training set size | | | | | | | | |
|----|-------|------|------|------|------|------|------|------|------|------|
| | | 10% | 20% | 30% | 40% | 50% | 60% | 70% | 80% | 90% |
| EX1 | GP | 0.02 ±0.0 | 0.02 ±0.0 | 0.02 ±0.0 | 0.02 ±0.0 | 0.02 ±0.0 | 0.02 ±0.0 | 0.02 ±0.0 | 0.02 ±0.0 | 0.02 ±0.0 |
| | k-NN | 0.04 ±0.01 | 0.02 ±0.0 | 0.02 ±0.0 | 0.02 ±0.0 | 0.02 ±0.0 | 0.02 ±0.0 | 0.02 ±0.0 | 0.02 ±0.0 | 0.02 ±0.0 |
| | RF | 0.02 ±0.0 | 0.02 ±0.0 | 0.02 ±0.0 | 0.02 ±0.0 | 0.02 ±0.0 | 0.02 ±0.0 | 0.02 ±0.0 | 0.02 ±0.0 | 0.02 ±0.0 |
| EX2 | GP | 0.01 ±0.0 | 0.01 ±0.0 | 0.01 ±0.0 | 0.01 ±0.0 | 0.01 ±0.0 | 0.01 ±0.0 | 0.01 ±0.0 | 0.01 ±0.0 | 0.01 ±0.0 |
| | k-NN | 0.02 ±0.0 | 0.01 ±0.0 | 0.01 ±0.0 | 0.01 ±0.0 | 0.01 ±0.0 | 0.01 ±0.0 | 0.01 ±0.0 | 0.01 ±0.0 | 0.01 ±0.0 |
| | RF | 0.01 ±0.0 | 0.01 ±0.0 | 0.01 ±0.0 | 0.01 ±0.0 | 0.01 ±0.0 | 0.01 ±0.0 | 0.01 ±0.0 | 0.01 ±0.0 | 0.01 ±0.0 |
| EX3 *adpcm* | GP | 0.04 ±0.03 | 0.01 ±0.0 | 0.01 ±0.0 | 0.01 ±0.0 | 0.01 ±0.0 | 0.01 ±0.0 | 0.01 ±0.0 | 0.01 ±0.0 | 0.01 ±0.0 |
| | k-NN | 0.06 ±0.01 | 0.03 ±0.01 | 0.02 ±0.01 | 0.02 ±0.01 | 0.01 ±0.01 | 0.01 ±0.0 | 0.01 ±0.0 | 0.01 ±0.0 | 0.01 ±0.0 |
| | RF | 0.01 ±0.0 | 0.01 ±0.0 | 0.01 ±0.0 | 0.01 ±0.0 | 0.01 ±0.0 | 0.01 ±0.0 | 0.01 ±0.0 | 0.01 ±0.0 | 0.01 ±0.0 |
| EX3 *aes* | GP | 0.01 ±0.0 | 0.01 ±0.0 | 0.0 ±0.0 | 0.0 ±0.0 | 0.0 ±0.0 | 0.0 ±0.0 | 0.0 ±0.0 | 0.0 ±0.0 | 0.0 ±0.0 |
| | k-NN | 0.05 ±0.02 | 0.03 ±0.01 | 0.01 ±0.0 | 0.0 ±0.0 | 0.0 ±0.0 | 0.0 ±0.0 | 0.0 ±0.0 | 0.0 ±0.0 | 0.0 ±0.0 |
| | RF | 0.0 ±0.0 | 0.0 ±0.0 | 0.0 ±0.0 | 0.0 ±0.0 | 0.0 ±0.0 | 0.0 ±0.0 | 0.0 ±0.0 | 0.0 ±0.0 | 0.0 ±0.0 |
| EX3 *dfadd* | GP | 0.04 ±0.01 | 0.03 ±0.01 | 0.03 ±0.01 | 0.03 ±0.0 | 0.03 ±0.0 | 0.02 ±0.0 | 0.02 ±0.0 | 0.02 ±0.0 | 0.02 ±0.0 |
| | k-NN | 0.11 ±0.04 | 0.05 ±0.01 | 0.04 ±0.01 | 0.03 ±0.01 | 0.03 ±0.0 | 0.03 ±0.0 | 0.03 ±0.0 | 0.03 ±0.0 | 0.03 ±0.0 |
| | RF | 0.05 ±0.01 | 0.03 ±0.01 | 0.03 ±0.01 | 0.03 ±0.0 | 0.03 ±0.0 | 0.02 ±0.0 | 0.02 ±0.0 | 0.02 ±0.0 | 0.02 ±0.0 |
| EX3 *dfmul* | GP | 0.04 ±0.01 | 0.03 ±0.01 | 0.02 ±0.0 | 0.02 ±0.0 | 0.02 ±0.0 | 0.02 ±0.0 | 0.02 ±0.0 | 0.02 ±0.0 | 0.02 ±0.0 |
| | k-NN | 0.11 ±0.03 | 0.05 ±0.01 | 0.03 ±0.01 | 0.02 ±0.0 | 0.03 ±0.01 | 0.02 ±0.0 | 0.02 ±0.0 | 0.02 ±0.0 | 0.02 ±0.0 |
| | RF | 0.05 ±0.02 | 0.03 ±0.01 | 0.02 ±0.0 | 0.02 ±0.0 | 0.02 ±0.0 | 0.02 ±0.0 | 0.02 ±0.0 | 0.02 ±0.0 | 0.02 ±0.0 |
| EX3 *gsm* | GP | 0.05 ±0.04 | 0.02 ±0.01 | 0.02 ±0.0 | 0.01 ±0.0 | 0.01 ±0.0 | 0.01 ±0.0 | 0.01 ±0.0 | 0.01 ±0.0 | 0.01 ±0.0 |
| | k-NN | 0.07 ±0.01 | 0.04 ±0.01 | 0.02 ±0.0 | 0.02 ±0.0 | 0.02 ±0.0 | 0.02 ±0.0 | 0.02 ±0.0 | 0.02 ±0.0 | 0.02 ±0.0 |
| | RF | 0.02 ±0.0 | 0.02 ±0.0 | 0.02 ±0.0 | 0.01 ±0.0 | 0.01 ±0.0 | 0.01 ±0.0 | 0.01 ±0.0 | 0.01 ±0.0 | 0.01 ±0.0 |
| EX3 *sha3* | GP | 0.19 ±0.04 | 0.15 ±0.02 | 0.15 ±0.01 | 0.14 ±0.01 | 0.14 ±0.01 | 0.14 ±0.01 | 0.13 ±0.01 | 0.13 ±0.01 | 0.13 ±0.01 |
| | k-NN | 0.19 ±0.02 | 0.16 ±0.02 | 0.16 ±0.02 | 0.15 ±0.02 | 0.15 ±0.02 | 0.16 ±0.02 | 0.16 ±0.02 | 0.15 ±0.02 | 0.15 ±0.02 |
| | RF | 0.2 ±0.03 | 0.17 ±0.03 | 0.16 ±0.02 | 0.15 ±0.02 | 0.14 ±0.02 | 0.14 ±0.02 | 0.13 ±0.02 | 0.14 ±0.02 | 0.14 ±0.02 |
| EX4 | GP | 0.25 ±0.01 | 0.25 ±0.01 | 0.24 ±0.01 | 0.24 ±0.01 | 0.24 ±0.01 | 0.24 ±0.01 | 0.24 ±0.01 | 0.24 ±0.01 | 0.24 ±0.01 |
| | RF | 0.25 ±0.01 | 0.25 ±0.01 | 0.24 ±0.01 | 0.24 ±0.01 | 0.24 ±0.01 | 0.24 ±0.01 | 0.24 ±0.01 | 0.24 ±0.01 | 0.24 ±0.01 |
| | k-NN | 0.26 ±0.02 | 0.25 ±0.02 | 0.25 ±0.01 | 0.25 ±0.01 | 0.25 ±0.01 | 0.26 ±0.02 | 0.25 ±0.01 | 0.26 ±0.01 | 0.26 ±0.02 |

Table 4: Normalized regression error and confidence interval for NX1 to NX9 scenarios. Since precision has been set to 0.01, results lower than this figure are shown as 0.0.

| NX | Model | Training set size | | | | | | | | |
|----|-------|------|------|------|------|------|------|------|------|------|
| | | 10% | 20% | 30% | 40% | 50% | 60% | 70% | 80% | 90% |
| NX1 | GP | 0.13 ±0.04 | 0.08 ±0.03 | 0.05 ±0.02 | 0.05 ±0.02 | 0.04 ±0.02 | 0.04 ±0.02 | 0.03 ±0.02 | 0.03 ±0.01 | 0.03 ±0.01 |
| | k-NN | 0.18 ±0.04 | 0.12 ±0.03 | 0.1 ±0.03 | 0.09 ±0.03 | 0.08 ±0.03 | 0.07 ±0.02 | 0.06 ±0.02 | 0.06 ±0.02 | 0.05 ±0.02 |
| | RF | 0.04 ±0.02 | 0.03 ±0.01 | 0.03 ±0.01 | 0.03 ±0.01 | 0.03 ±0.01 | 0.03 ±0.01 | 0.03 ±0.01 | 0.02 ±0.01 | 0.02 ±0.01 |
| NX2 | GP | 0.39 ±0.08 | 0.32 ±0.06 | 0.29 ±0.06 | 0.28 ±0.04 | 0.27 ±0.04 | 0.27 ±0.04 | 0.26 ±0.03 | 0.26 ±0.03 | 0.26 ±0.03 |
| | k-NN | 0.36 ±0.07 | 0.32 ±0.05 | 0.3 ±0.05 | 0.3 ±0.05 | 0.29 ±0.05 | 0.28 ±0.05 | 0.27 ±0.05 | 0.26 ±0.04 | 0.26 ±0.04 |
| | RF | 0.3 ±0.05 | 0.27 ±0.03 | 0.27 ±0.04 | 0.27 ±0.04 | 0.27 ±0.04 | 0.26 ±0.04 | 0.26 ±0.04 | 0.26 ±0.04 | 0.26 ±0.04 |
| NX3 | GP | 0.14 ±0.04 | 0.09 ±0.03 | 0.07 ±0.02 | 0.05 ±0.02 | 0.05 ±0.02 | 0.04 ±0.01 | 0.04 ±0.01 | 0.04 ±0.01 | 0.04 ±0.01 |
| | k-NN | 0.18 ±0.03 | 0.13 ±0.02 | 0.1 ±0.02 | 0.08 ±0.02 | 0.07 ±0.02 | 0.07 ±0.02 | 0.06 ±0.01 | 0.06 ±0.02 | 0.06 ±0.01 |
| | RF | 0.04 ±0.01 | 0.04 ±0.0 | 0.03 ±0.01 | 0.03 ±0.01 | 0.03 ±0.0 | 0.03 ±0.0 | 0.03 ±0.0 | 0.03 ±0.0 | 0.03 ±0.0 |
| NX4 | GP | 0.26 ±0.05 | 0.21 ±0.02 | 0.19 ±0.03 | 0.19 ±0.02 | 0.18 ±0.03 | 0.17 ±0.02 | 0.17 ±0.02 | 0.17 ±0.03 | 0.17 ±0.02 |
| | k-NN | 0.29 ±0.03 | 0.24 ±0.04 | 0.22 ±0.04 | 0.2 ±0.03 | 0.19 ±0.03 | 0.19 ±0.03 | 0.19 ±0.03 | 0.18 ±0.03 | 0.19 ±0.03 |
| | RF | 0.2 ±0.03 | 0.19 ±0.03 | 0.18 ±0.03 | 0.18 ±0.03 | 0.17 ±0.03 | 0.17 ±0.03 | 0.17 ±0.03 | 0.17 ±0.03 | 0.17 ±0.03 |
| NX5 | GP | 0.22 ±0.04 | 0.18 ±0.03 | 0.16 ±0.02 | 0.15 ±0.02 | 0.15 ±0.02 | 0.14 ±0.02 | 0.13 ±0.02 | 0.14 ±0.02 | 0.13 ±0.02 |
| | k-NN | 0.26 ±0.04 | 0.2 ±0.03 | 0.18 ±0.03 | 0.17 ±0.03 | 0.17 ±0.02 | 0.16 ±0.02 | 0.15 ±0.02 | 0.15 ±0.02 | 0.15 ±0.02 |
| | RF | 0.16 ±0.02 | 0.14 ±0.01 | 0.14 ±0.02 | 0.13 ±0.02 | 0.13 ±0.01 | 0.13 ±0.01 | 0.13 ±0.01 | 0.12 ±0.02 | 0.13 ±0.01 |
| NX6 | GP | 0.15 ±0.04 | 0.09 ±0.02 | 0.06 ±0.01 | 0.05 ±0.01 | 0.05 ±0.01 | 0.04 ±0.01 | 0.04 ±0.01 | 0.04 ±0.01 | 0.04 ±0.01 |
| | k-NN | 0.21 ±0.03 | 0.14 ±0.02 | 0.11 ±0.02 | 0.09 ±0.02 | 0.08 ±0.01 | 0.07 ±0.01 | 0.06 ±0.01 | 0.06 ±0.01 | 0.05 ±0.01 |
| | RF | 0.07 ±0.01 | 0.05 ±0.01 | 0.04 ±0.01 | 0.04 ±0.0 | 0.04 ±0.0 | 0.04 ±0.0 | 0.03 ±0.0 | 0.03 ±0.0 | 0.03 ±0.0 |
| NX7 | GP | 0.09 ±0.0 | 0.09 ±0.0 | 0.09 ±0.0 | 0.09 ±0.0 | 0.09 ±0.0 | 0.09 ±0.0 | 0.09 ±0.0 | 0.09 ±0.0 | 0.09 ±0.0 |
| | RF | 0.09 ±0.0 | 0.09 ±0.0 | 0.09 ±0.0 | 0.09 ±0.0 | 0.09 ±0.0 | 0.09 ±0.0 | 0.09 ±0.0 | 0.09 ±0.0 | 0.09 ±0.0 |
| | k-NN | 0.1 ±0.01 | 0.1 ±0.01 | 0.1 ±0.01 | 0.1 ±0.01 | 0.1 ±0.01 | 0.1 ±0.01 | 0.1 ±0.01 | 0.09 ±0.0 | 0.1 ±0.01 |
| NX8 | GP | 0.02 ±0.0 | 0.02 ±0.0 | 0.02 ±0.0 | 0.02 ±0.0 | 0.02 ±0.0 | 0.02 ±0.0 | 0.02 ±0.0 | 0.02 ±0.0 | 0.02 ±0.0 |
| | RF | 0.02 ±0.0 | 0.02 ±0.0 | 0.02 ±0.0 | 0.02 ±0.0 | 0.02 ±0.0 | 0.02 ±0.0 | 0.02 ±0.0 | 0.02 ±0.0 | 0.02 ±0.0 |
| | k-NN | 0.03 ±0.0 | 0.03 ±0.0 | 0.03 ±0.0 | 0.03 ±0.0 | 0.02 ±0.0 | 0.03 ±0.0 | 0.03 ±0.0 | 0.02 ±0.0 | 0.02 ±0.0 |
| NX9 | GP | 0.01 ±0.0 | 0.01 ±0.0 | 0.01 ±0.0 | 0.01 ±0.0 | 0.01 ±0.0 | 0.01 ±0.0 | 0.01 ±0.0 | 0.01 ±0.0 | 0.01 ±0.0 |
| | RF | 0.01 ±0.0 | 0.01 ±0.0 | 0.01 ±0.0 | 0.01 ±0.0 | 0.01 ±0.0 | 0.01 ±0.0 | 0.01 ±0.0 | 0.01 ±0.0 | 0.01 ±0.0 |
| | k-NN | 0.01 ±0.0 | 0.01 ±0.0 | 0.01 ±0.0 | 0.01 ±0.0 | 0.01 ±0.0 | 0.01 ±0.0 | 0.01 ±0.0 | 0.01 ±0.0 | 0.01 ±0.0 |

Table 5: Normalized regression error and confidence interval for NX10 to NX18 scenarios. Since precision has been set to 0.01, results lower than this figure are shown as 0.0.

| NX | Model | Training set size | | | | | | | | |
|---|---|---|---|---|---|---|---|---|---|---|
| | | 10% | 20% | 30% | 40% | 50% | 60% | 70% | 80% | 90% |
| NX10 | GP | 0.05 ±0.0 | 0.04 ±0.0 | 0.04 ±0.0 | 0.04 ±0.0 | 0.04 ±0.0 | 0.04 ±0.0 | 0.04 ±0.0 | 0.04 ±0.0 | 0.04 ±0.0 |
| | RF | 0.05 ±0.0 | 0.04 ±0.0 | 0.04 ±0.0 | 0.04 ±0.0 | 0.04 ±0.0 | 0.04 ±0.0 | 0.04 ±0.0 | 0.04 ±0.0 | 0.04 ±0.0 |
| | k-NN | 0.05 ±0.01 | 0.05 ±0.01 | 0.05 ±0.0 | 0.05 ±0.01 | 0.05 ±0.0 | 0.05 ±0.0 | 0.05 ±0.0 | 0.05 ±0.0 | 0.05 ±0.0 |
| NX11 | GP | 0.04 ±0.01 | 0.04 ±0.0 | 0.04 ±0.0 | 0.03 ±0.0 | 0.03 ±0.0 | 0.03 ±0.0 | 0.03 ±0.0 | 0.03 ±0.0 | 0.03 ±0.0 |
| | RF | 0.04 ±0.0 | 0.03 ±0.0 | 0.03 ±0.0 | 0.03 ±0.0 | 0.03 ±0.0 | 0.03 ±0.0 | 0.03 ±0.0 | 0.03 ±0.0 | 0.03 ±0.0 |
| | k-NN | 0.06 ±0.01 | 0.05 ±0.01 | 0.04 ±0.01 | 0.04 ±0.01 | 0.04 ±0.01 | 0.04 ±0.01 | 0.04 ±0.0 | 0.04 ±0.0 | 0.04 ±0.0 |
| NX12 | GP | 0.03 ±0.0 | 0.03 ±0.0 | 0.03 ±0.0 | 0.03 ±0.0 | 0.03 ±0.0 | 0.03 ±0.0 | 0.03 ±0.0 | 0.03 ±0.0 | 0.03 ±0.0 |
| | RF | 0.03 ±0.0 | 0.03 ±0.0 | 0.03 ±0.0 | 0.03 ±0.0 | 0.03 ±0.0 | 0.03 ±0.0 | 0.03 ±0.0 | 0.03 ±0.0 | 0.03 ±0.0 |
| | k-NN | 0.03 ±0.0 | 0.03 ±0.0 | 0.03 ±0.0 | 0.03 ±0.0 | 0.03 ±0.0 | 0.03 ±0.0 | 0.03 ±0.0 | 0.03 ±0.0 | 0.03 ±0.0 |
| NX13 | GP | 0.05 ±0.0 | 0.05 ±0.0 | 0.05 ±0.0 | 0.05 ±0.0 | 0.05 ±0.0 | 0.05 ±0.0 | 0.05 ±0.0 | 0.05 ±0.0 | 0.05 ±0.0 |
| | RF | 0.05 ±0.0 | 0.05 ±0.0 | 0.05 ±0.0 | 0.05 ±0.0 | 0.05 ±0.0 | 0.05 ±0.0 | 0.05 ±0.0 | 0.05 ±0.0 | 0.05 ±0.0 |
| | k-NN | 0.05 ±0.0 | 0.05 ±0.0 | 0.05 ±0.0 | 0.05 ±0.0 | 0.05 ±0.0 | 0.05 ±0.0 | 0.06 ±0.0 | 0.05 ±0.0 | 0.05 ±0.0 |
| NX14 | GP | 0.04 ±0.0 | 0.04 ±0.0 | 0.04 ±0.0 | 0.04 ±0.0 | 0.04 ±0.0 | 0.04 ±0.0 | 0.04 ±0.0 | 0.04 ±0.0 | 0.04 ±0.0 |
| | RF | 0.04 ±0.0 | 0.04 ±0.0 | 0.04 ±0.0 | 0.04 ±0.0 | 0.04 ±0.0 | 0.04 ±0.0 | 0.04 ±0.0 | 0.04 ±0.0 | 0.04 ±0.0 |
| | k-NN | 0.05 ±0.0 | 0.04 ±0.0 | 0.04 ±0.0 | 0.04 ±0.0 | 0.04 ±0.0 | 0.04 ±0.0 | 0.04 ±0.0 | 0.04 ±0.0 | 0.04 ±0.0 |
| NX15 | GP | 0.06 ±0.0 | 0.06 ±0.0 | 0.06 ±0.0 | 0.06 ±0.0 | 0.06 ±0.0 | 0.06 ±0.0 | 0.06 ±0.0 | 0.06 ±0.0 | 0.06 ±0.0 |
| | RF | 0.06 ±0.0 | 0.06 ±0.0 | 0.06 ±0.0 | 0.06 ±0.0 | 0.06 ±0.0 | 0.06 ±0.0 | 0.06 ±0.0 | 0.06 ±0.0 | 0.06 ±0.0 |
| | k-NN | 0.07 ±0.0 | 0.07 ±0.0 | 0.07 ±0.0 | 0.06 ±0.0 | 0.07 ±0.0 | 0.07 ±0.0 | 0.07 ±0.0 | 0.07 ±0.0 | 0.07 ±0.0 |
| NX16 | GP | 0.03 ±0.0 | 0.03 ±0.0 | 0.02 ±0.0 | 0.02 ±0.0 | 0.02 ±0.0 | 0.02 ±0.0 | 0.02 ±0.0 | 0.02 ±0.0 | 0.02 ±0.0 |
| | RF | 0.02 ±0.0 | 0.02 ±0.0 | 0.02 ±0.0 | 0.02 ±0.0 | 0.02 ±0.0 | 0.02 ±0.0 | 0.02 ±0.0 | 0.02 ±0.0 | 0.02 ±0.0 |
| | k-NN | 0.03 ±0.0 | 0.03 ±0.0 | 0.03 ±0.0 | 0.03 ±0.0 | 0.03 ±0.0 | 0.03 ±0.0 | 0.03 ±0.0 | 0.03 ±0.0 | 0.03 ±0.0 |
| NX17 | GP | 0.03 ±0.0 | 0.03 ±0.0 | 0.03 ±0.0 | 0.03 ±0.0 | 0.03 ±0.0 | 0.03 ±0.0 | 0.03 ±0.0 | 0.03 ±0.0 | 0.03 ±0.0 |
| | RF | 0.03 ±0.0 | 0.03 ±0.0 | 0.03 ±0.0 | 0.03 ±0.0 | 0.03 ±0.0 | 0.03 ±0.0 | 0.03 ±0.0 | 0.03 ±0.0 | 0.03 ±0.0 |
| | k-NN | 0.03 ±0.0 | 0.03 ±0.0 | 0.03 ±0.0 | 0.03 ±0.0 | 0.03 ±0.0 | 0.03 ±0.0 | 0.03 ±0.0 | 0.03 ±0.0 | 0.03 ±0.0 |
| NX18 | GP | 0.58 ±0.15 | 0.44 ±0.12 | 0.37 ±0.1 | 0.35 ±0.1 | 0.31 ±0.08 | 0.3 ±0.07 | 0.28 ±0.07 | 0.27 ±0.07 | 0.27 ±0.07 |
| | RF | 0.33 ±0.09 | 0.27 ±0.05 | 0.27 ±0.06 | 0.24 ±0.03 | 0.24 ±0.03 | 0.23 ±0.03 | 0.23 ±0.03 | 0.22 ±0.03 | 0.22 ±0.03 |
| | k-NN | 0.27 ±0.02 | 0.24 ±0.03 | 0.24 ±0.03 | 0.24 ±0.03 | 0.23 ±0.02 | 0.23 ±0.03 | 0.23 ±0.03 | 0.23 ±0.02 | 0.22 ±0.02 |

Table 6: Normalized regression error and confidence interval for NX19 and NX20 scenarios. Since precision has been set to 0.01, results lower than this figure are shown as 0.0.

| NX | Model | Training set size | | | | | | | | |
|---|---|---|---|---|---|---|---|---|---|---|
| | | 10% | 20% | 30% | 40% | 50% | 60% | 70% | 80% | 90% |
| NX19 | GP | 0.14 ±0.02 | 0.13 ±0.02 | 0.12 ±0.01 | 0.12 ±0.01 | 0.12 ±0.01 | 0.12 ±0.01 | 0.12 ±0.01 | 0.12 ±0.01 | 0.12 ±0.01 |
| | RF | 0.14 ±0.02 | 0.13 ±0.01 | 0.12 ±0.01 | 0.12 ±0.01 | 0.12 ±0.01 | 0.12 ±0.01 | 0.12 ±0.01 | 0.12 ±0.01 | 0.12 ±0.01 |
| | k-NN | 0.15 ±0.02 | 0.13 ±0.02 | 0.13 ±0.02 | 0.13 ±0.02 | 0.13 ±0.01 | 0.13 ±0.02 | 0.13 ±0.02 | 0.13 ±0.01 | 0.13 ±0.02 |
| NX20 | GP | 0.11 ±0.01 | 0.11 ±0.01 | 0.11 ±0.01 | 0.1 ±0.01 | 0.1 ±0.01 | 0.1 ±0.01 | 0.1 ±0.01 | 0.1 ±0.01 | 0.1 ±0.01 |
| | RF | 0.11 ±0.01 | 0.11 ±0.01 | 0.1 ±0.01 | 0.1 ±0.01 | 0.1 ±0.01 | 0.1 ±0.01 | 0.1 ±0.01 | 0.1 ±0.01 | 0.1 ±0.01 |
| | k-NN | 0.13 ±0.01 | 0.12 ±0.01 | 0.11 ±0.01 | 0.11 ±0.01 | 0.11 ±0.01 | 0.11 ±0.01 | 0.11 ±0.01 | 0.11 ±0.01 | 0.11 ±0.01 |

