# OpenReview forum: "TANGRAM: A Dataset for FPGA-Based Heterogeneous Systems-on-Chip Optimization"
_ICLR.cc/2026/Conference — Submitted to ICLR 2026_

### Official Review · Reviewer_xqCw · 2025-10-31

**Soundness:** 2
**Presentation:** 3
**Contribution:** 2
**Rating:** 4
**Confidence:** 3

**Summary:**

The authors introduce TANGRAM, a dataset comprising performance (throughput) and resource utilization (area) statistics for over 40,000 unique heterogeneous System-on-Chip (SoC) designs. This data was collected via resource-intensive hardware implementation and execution on two high-end FPGA platforms). The paper posits that this dataset is crucial for enabling ML-based design space exploration (DSE) for FPGAs, a task currently dominated by suboptimal heuristics. To demonstrate TANGRAM's utility, the authors provide baseline results for two ML tasks: 1) a regression task to predict accelerator throughput based on design features, using models like Gaussian Processes and Random Forest, and 2) an optimization task using a multi-armed bandit (MAB) approach to find the minimal-area design that meets specific performance (throughput) constraints.

**Strengths:**

(1) Real hardware measurements (not only simulation) on two high-end boards — this is costly and slow, so packaging it is valuable.
(2) The paper clearly reports what is exhaustive and what is non-exhaustive (Tables 1–2), so users know where ground truth is complete.
(3) Baseline ML runs are reproducible with code and data provided in the supplementary material.

**Weaknesses:**

(1) Only two FPGA families and one style of SoC are involved. Thus, it is not obvious models trained on this will generalize to other NoCs, to other FPGA vendors, or to MPSoCs with heterogeneous interconnects. Right now the dataset mostly tells us “how this open-source platform behaves”. That is good engineering, but limited research scope.
(2) The regression task uses only 3 features per accelerator: parallelism level, #memory-bound co-runners, #compute-bound co-runners, which seems a rather coarse parametrization of the problem. Critical features are ignored: accelerator-specific characteristics, placement and routing related features, etc.
(3) The multi-armed bandit (MAB) optimization is only tested on the small, exhaustive (EX) scenarios. The paper's motivation is to solve problems where the design space is too large to explore exhaustively (i.e., the non-exhaustive (NX) scenarios). Validating an optimization algorithm on a tiny dataset where the full ground truth is already known is unrealistic and feels like a strawman argument.

**Questions:**

(1) At what error rate does the dataset become impractical for real design decisions? Can you demonstrate end-to-end SoC design using your dataset where ML predictions led to a near-optimal configuration?
(2) Can you experiment and compare against a Transformer/GNN-based method? For ICLR, the audience might in general expect at least a graph- or set-based baseline to show that the dataset actually benefits from structure-aware models.

---

### Official Review · Reviewer_WcV5 · 2025-10-31

**Soundness:** 2
**Presentation:** 2
**Contribution:** 1
**Rating:** 2
**Confidence:** 4

**Summary:**

This paper introduces TANGRAM, a curated dataset of >40k heterogeneous SoC configurations measured on real FPGA hardware (Virtex‑7 2000T and Alveo U55C with HBM2). Each configuration specifies a 2D‑mesh NoC layout, tile assignments (CPU/AUX/MEM/ACC/EMPTY), accelerator parallelism, and reports area (BRAM18/DSP/FF/LUT) and throughput both per‑tile and aggregated per application. The dataset spans exhaustive settings for small design spaces (EX1-EX4; 5.5k configs) and non‑exhaustive settings for larger spaces (NX1-NX20; 37k configs), with selections obtained by random sampling and a bandit‑style selector. The paper validates dataset utility with baseline multi‑output regression (GP, RF, k-NN) using very compact features (the accelerator’s parallelism and counts of memory‑bound vs compute‑bound peers on the same memory tile) and a combinatorial multi‑armed bandit (FC‑MAB) for area‑minimizing design selection under throughput constraints. Representative plots show that RF/GP achieve low prediction error in some scenarios (e.g., EX3), while the bandit can reach the optimal solution on EX4 after exploring around 1.5% of the space.

**Strengths:**

A large, hardware‑measured dataset at the SoC‑configuration level is rare and valuable for ML‑for‑systems research. Unlike simulated corpora, TANGRAM reports real board measurements across two FPGA families, capturing contention effects that simple performance models miss.

The paper documents exhaustive (EX1-EX4) and non‑exhaustive (NX1-NX20) coverage with clear scenario cards (NoC sizes, app sets, memory type) totals around 42.7k configs.

The paper/idea has a clear schema: JSON entries include area, per‑accelerator and total throughput, and tile coordinates. This is straightforward to parse and extend.

**Weaknesses:**

The regression uses very limited features - parallelism and coarse counts of memory‑/compute‑bound peers. No explicit structural/topological features (e.g., NoC hop distances to MEM tiles, link bisection bandwidth, contention proxies), nor per‑accelerator characteristics beyond a categorical app label are used. This likely explains persistent high errors in harder settings (e.g., EX4 plateaus near ~0.24 even at 90% training in Table 3, p.15; NX18 ~0.22–0.58). The paper’s headline “accurate predictions” claim is supported for EX3‑like cases but is not uniformly strong across scenarios with HBM2 or larger NoCs. The paper should: (i) add topology‑aware features, or (ii) provide results with graph neural networks or learned embeddings of tile layouts to demonstrate the dataset’s ML headroom.

All regressions appear intra‑scenario; the paper does not study cross‑scenario generalization (e.g., train on Virtex‑7 BRAM, test on UltraScale+ HBM2; or train on 3$\times$3, test on 4$\times$4). Without this, it is hard to judge the dataset’s utility for zero/low‑shot deployment on new devices - arguably the central appeal of ML vs. heuristics highlighted in Section 3.

The optimization baseline is limited. The FC‑MAB comparison is only against a random selector and only on exhaustive scenarios where the true optimum is known. There is no comparison to (a) heuristics from prior work (e.g., MILP surrogates, genetic knapsack, HPO‑style DSE cited in Section 3), or (b) BO/Bayesian optimization with surrogate models, or (c) simple greedy/ILP with learned cost. Actionable: add at least one strong, reproducible baseline beyond RAND (e.g., RF‑surrogate+EI BO, or a genetic algorithm), and run in a non‑exhaustive setting leveraging only the available subset to mimic the realistic use case.

Execution methodology says accelerators run "for several seconds," sending stats over UART (Section 4.1), but run‑to‑run variance, warm‑up effects, clocking/DVFS, and measurement repeatability are not quantified. Confidence intervals in regression reflect model CV, not hardware measurement noise.

Some figures (e.g., Figure 5,) aggregate errors across apps without per‑app breakdowns; per‑app plots in the appendix tables help but compact per‑scenario per‑app plots would clarify where models fail.

Units/definitions (e.g., MB/s vs MiB/s, “memory‑/compute‑bound” classifier source) could be stated more precisely.

Overall, this paper does not seem to be well suited for the ICLR conference: it seems that it would be better suited at a hardware type conference.

**Questions:**

Can you report train on BRAM / test on HBM2 (and vice versa), and 3×3→4×4 NoC generalization? Even a modest transfer experiment would substantiate the dataset’s cross‑domain utility.

Do you have preliminary results using graph encodings of the tile layout (e.g., per‑tile features + NoC edges) with a GNN or transformer encoder? This seems well‑aligned with your Figure 1 setup.

How many measurement repetitions per configuration? Can you publish per‑config confidence intervals or raw time series to quantify noise and enable probabilistic models?

---

### Official Review · Reviewer_NFdE · 2025-11-01

**Soundness:** 3
**Presentation:** 2
**Contribution:** 2
**Rating:** 4
**Confidence:** 5

**Summary:**

This paper presents TANGRAM, a large, open dataset of real FPGA-based heterogeneous SoC configurations, targeting ML-driven modeling and design-space exploration. Configurations place specialized accelerator tiles on a 2D-mesh NoC alongside CPU and memory tiles; the dataset spans multiple scenarios across Virtex-7/US, totaling tens of thousands of configurations. Given the vast design space in SoC optimization, TANGRAM provides ground-truth area/performance to enable accurate prediction and efficient search. The paper validates utility with two ML tasks: multi-output regression to predict throughput from configuration features, and a combinatorial multi-armed bandit to minimize area under throughput SLAs, showing <5% error with 20 – 30% training data and near-optimal configurations after exploring ~1.5% of the design space. TANGRAM is a benchmark for ML-for-systems research on learned performance modeling and data-driven SoC DSE.

The key contribution of this paper is an FPGA + ML for Design Space Exploration (DSE) dataset. While the dataset can be impactful for the ML for DSE, system optimization communities, the lack of any insights into learning representations makes it suspect for ICLR as a venue. As an applied ML + DSE on FPGA paper with little connection to insights into learning representations, it would be more suitable for a ML systems conference (MLSys) or an FPGA conference (FPGA,FPL, FCCM) which evaluate ML workloads on FPGA.

**Strengths:**

- The paper provides a novel dataset and scope. TANGRAM is the first large-scale, open-source dataset capturing real FPGA-SoC implementations with heterogeneous accelerator placement, area, and throughput statistics across tens of thousands of configurations. This fills a longstanding gap between purely simulated benchmarks (e.g., HLSynth, IronMan) and realistic hardware measurements.

- The paper has high engineering and reproducibility quality. The data collection methodology is rigorous – every configuration is synthesized and executed on physical FPGA hardware, and the dataset (JSON-based) is released with full scripts and schema for replication and extension.

- By framing throughput prediction and area-throughput optimization as regression and bandit-learning tasks, the paper shows that the dataset can serve as a benchmark for ML-guided design-space exploration (ML4EDA / ML4Systems).

- The empirical validation seems sound. Regression models achieve <5% normalized error using 20 – 30% training data, while the multi-armed bandit finds near-optimal SoC configurations after exploring only ~1.5% of the search space – demonstrating real potential for learning-based design acceleration.

- Presentation and documentation is good. The paper is clearly written, well-structured, and includes transparent figures, examples, and dataset description (JSON schema, feature definitions, and reproducibility details.

- TANGRAM provides a foundational dataset for future ML research on FPGA/SoC optimization, enabling new work on performance modeling and reinforcement-learning-driven placement strategies.

**Weaknesses:**

There are several weaknesses primarily related to new insights into learning (representations) as well as some empirical evaluation limitations that make this les relevant to ICLR and more of a ML systems or FPGA related work These are:

- Limited learning innovation: The paper primarily contributes a dataset and evaluation infrastructure rather than new learning methods. The regression and multi-armed bandit baselines are standard and serve only as proof-of-concept demonstrations, limiting methodological depth.

- The modeling setup is quite simplistic. The regression task relies on a few handcrafted features (e.g., accelerator parallelism and memory-sharing counts). Given the dataset’s structured 2D topology, richer models such as graph neural networks could more effectively capture inter-tile dependencies and spatial correlations.

- There is a lack of architectural diversity in the configurations considered in the paper. The dataset currently focuses on 2D-mesh NoC configurations with a relatively small number of accelerators. Broader inclusion of different interconnect topologies, heterogeneous memory systems, or larger SoC scales would strengthen its generality and long-term value.

- Evaluation quality is compromised by a lack of comparative baselines. While related frameworks such as IronMan-Pro and MLNoC are cited, no cross-dataset evaluation is conducted to quantitatively establish the realism or transferability of TANGRAM’s measurements.

- Benchmark framing could be stronger. While the paper shows utility through regression and optimization tasks, it does not yet define canonical benchmark challenges or standardized splits that would help future researchers adopt and extend the dataset consistently.

The key contribution of this paper is an FPGA + ML for Design Space Exploration (DSE) dataset. While the dataset can be impactful for the ML for DSE, system optimization communities, the lack of any insights into learning representations makes it suspect for ICLR as a venue. As an applied ML + DSE on FPGA paper with little connection to insights into learning representations, it would be more suitable for a ML systems conference (MLSys) or an FPGA conference (FPGA,FPL, FCCM) which evaluate ML workloads on FPGA. Technically: the dataset does not come up with a set of representative tasks -- they do supply two tasks related to throughput prediction and area minimization but its not really standardized in any way which would question the extent of its utility/applicability to ML community working in the space.

**Questions:**

Please see weakness comments as well in addition to the following technical questions.

- How flexible is the current JSON schema in supporting alternative NoC topologies (e.g., torus, tree, bus) or different tile types? Could the same schema generalize to emerging FPGA-SoC fabrics such as Versal ACAP?

- For the “non-exhaustive” subsets, what guided the sampling of configurations? Were these selected randomly or using domain heuristics (e.g., balance of compute- vs memory-bound workloads)? This lack of detail makes it hard to make conclusions on generalizability.

- Were post-route timing, routing congestion, or thermal effects included in the recorded metrics, or are results based solely on functional execution? How significant are runtime variations across reconfigurations of the same design?

- Why are you using only three scalar features (parallelism, number of memory- and compute-bound accelerators) for regression? Did you experiment with topology-aware or spatial features, and if so, how did those affect prediction accuracy?

- How consistent are the performance trends across the Virtex-7 and U55C datasets? Is there evidence that models trained on one platform generalize to the other?

- Will future versions of TANGRAM expand to include larger designs or additional FPGA architectures?

- The multi-armed bandit experiment explores ~1.5 % of the design space – what specific reward formulation and convergence criterion were used, and how sensitive are the results to these parameters?

- Have you considered defining canonical ML tasks – e.g., graph-based placement prediction or resource-aware performance modeling – on top of TANGRAM to formalize it as a benchmark suite?

---

### Official Review · Reviewer_VTDQ · 2025-11-01

**Soundness:** 2
**Presentation:** 2
**Contribution:** 2
**Rating:** 2
**Confidence:** 5

**Summary:**

This paper introduces TANGRAM, a large-scale dataset designed for the performance and area characterization of heterogeneous FPGA-based Systems-on-Chip (SoCs). The dataset includes over 40,000 SoC configurations, capturing detailed metrics such as resource utilization (LUTs, BRAMs, DSPs, and FFs) and throughput across multiple accelerators and applications. The authors use two high-end FPGA platforms (AMD Virtex-7 and Alveo U55C) to gather real execution data rather than simulated results, enhancing realism and reliability.

**Strengths:**

The paper makes a practical contribution by proposing a comprehensive SoC dataset that includes diverse configurations along with their performance and area metrics.

**Weaknesses:**

1. The dataset focuses on two FPGA families and a narrow range of NoC and application configurations, limiting its applicability to broader SoC or ASIC design domains. The paper does not convincingly show how well models trained on TANGRAM would transfer to other architectures or workloads.
2. The dataset omits key workloads like GEMM that dominate modern deep learning applications, and its reliance on CHStone kernels further restricts the relevance to today's ML-driven optimization problems.
3. The use of relatively small 2×3 and 3×3 meshes constrains scalability and may not capture the complex communication patterns found in larger or hierarchical NoCs used in production SoCs.
4. The dataset primarily features legacy or synthetic benchmarks, missing compute-intensive workloads like convolution, matrix multiplication, or transformer-based kernels that are central to current AI accelerators.
5. The experiments merely compare standard ML regressors and bandit algorithms without explaining why TANGRAM is uniquely valuable for learning-based SoC optimization.

**Questions:**

1. Fig. 2: Why is throughput reported in MB/s instead of OPs/s?
2. How would the proposed methodology handle architectures like AMD AI Engine, which are also NoC-based but include hierarchical memory and distributed control logic?
3. The current 3×3 and 4×4 mesh topologies are too small to demonstrate scaling behavior. Why not incorporate larger designs in your dataset?
4. The absence of GEMM, a fundamental building block for deep learning and HPC, weakens the dataset's applicability. Do you have results on GEMM or other neural network operators?
5. The CHStone benchmark suite is outdated and too small for evaluating ML-based design methods. More modern suites such as Rosetta [A] would offer broader coverage of real-world compute patterns.
6. The experiments should include comparisons with existing state-of-the-art HLS prediction or exploration frameworks (e.g., IronMan-Pro [B], AutoDSE [C], GNN-DSE [D]) to contextualize the dataset's usefulness.
7. Do you have results on using more advanced neural network-based models (e.g., graph neural networks or Transformer-based models) that can capture structural relationships between tiles and accelerators, which are central to SoC performance prediction?

[A] Zhou, Yuan, et al. "Rosetta: A realistic high-level synthesis benchmark suite for software programmable FPGAs." Proceedings of the 2018 ACM/SIGDA International Symposium on Field-Programmable Gate Arrays. 2018.

[B] Wu, Nan, Yuan Xie, and Cong Hao. "IRONMAN-PRO: Multiobjective design space exploration in HLS via reinforcement learning and graph neural network-based modeling." IEEE Transactions on Computer-Aided Design of Integrated Circuits and Systems 42.3 (2022): 900-913.

[C] Sohrabizadeh, Atefeh, et al. "AutoDSE: Enabling software programmers to design efficient FPGA accelerators." ACM Transactions on Design Automation of Electronic Systems (TODAES) 27.4 (2022): 1-27.

[D] Sohrabizadeh, Atefeh, et al. "Automated accelerator optimization aided by graph neural networks." Proceedings of the 59th ACM/IEEE Design Automation Conference. 2022.

---

### Meta-Review · Area_Chair_qkeD · 2026-01-04

**Summary:**

The paper had two rejects and two weak rejects. The authors **did not submit a rebuttal**. Given this is such a clear case, I recommend rejection.

**Reviewer Concerns:**

None of the concerns raised were addressed by the rebuttal, as there was no rebuttal.

**Reviewer Scores:**

Reviewers would not have changed their score if they had been able to participate fully in the discussion.

---

### Decision · Program_Chairs · 2026-01-26

Reject